# OpenReview forum: "On the Geometry of Regularization in Adversarial Training: High-Dimensional Asymptotics and Generalization Bounds"
_ICML.cc/2025/Conference — Submitted to ICML 2025_

### Official Review · Reviewer_jKDA · 2025-03-04

**Overall Recommendation:** 4

**Summary:**

This paper studies the effect of explicit regularization in adversarial training using sharp high-dimensional asymptotics. Its main qualitative result is that regularizing using the dual of the norm with respect to which the perturbation budget is defined yields significant performance gains.

## Update after rebuttal

After the rebuttal, my opinion remains positive.

**Claims And Evidence:**

All claims are well-supported.

**Essential References Not Discussed:**

The authors provide a satisfactory review of prior work.

**Experimental Designs Or Analyses:**

The experiments are adequate. I do wish the authors mentioned the choice of covariance structure used in Figure 3 in the main text rather than completely deferring that detail to Appendix C.

**Methods And Evaluation Criteria:**

The methods are appropriate.

**Other Comments Or Suggestions:**

- In Line 165, "euclidean" -> "Euclidean"

- For the reader's convenience, it would be nice to link to the appendix containing the relevant proofs after each theorem.

- I would suggest that the authors consider swapping their use of color and dashing in Figure 3; I think using different colors for different norms and different dashing for different $\epsilon$ would be more legible. Alternatively, they could use different colors for different $\epsilon$ and differentiate norms by the saturation of the colors. The dotted and dashed lines are too similar to be optimally legible.

- In Line 862, there is a jumbled reference to Loureiro et al. 2021: "(?)Lemma 11]loureiro2021learning".

**Other Strengths And Weaknesses:**

- All experiments in the submitted manuscript use Gaussian data. As the authors do not have a proof of universality for the problem at hand, I think an experiment showing that the main conceptual result (i.e., regularizing with the dual norm improves performance) transfers to a real dataset would enhance the paper. However, I leave this to the authors' discretion, as I think the paper could be accepted in its submitted form.

- I wish the authors were able to extract more insight directly from Theorem 3.15. I fully appreciate that these self-consistent equations are not easily analyzable, but Remark 3.17 doesn't really say anything specific to this result. It merely restates the standard idea that the order parameters are sufficient statistics, and mentions the proof strategy (which is re-stated again in lines 297-301). Are there any special cases (beyond isotropic data) for which you could make something more of this result?

- I think more could be done to integrate the Rademacher complexity analysis in Section 4 with the sharp asymptotics derived in the preceding sections. As it stands, the reader is left feeling that the main conceptual message comes from this analysis, which then could be supported through numerics alone rather than through numerical solution of the self-consistent equations giving the sharp asymptotic. I acknowledge that a complete analysis of the transition to optimality of the dual norm with increasing perturbation scale is out of reach, but (in the same vein as my previous comment) I wish the authors showed how more can be done with access to the sharp asymptotic. At the very least, the authors should show how loose the Rademacher complexity bounds are.

**Questions For Authors:**

Given the smooth transition between $r^{\star} = 2$ and $r^{\star} = 1$ with increasing $\epsilon$ that you observe in Figure 4, can you gain any insight by expanding around $\epsilon = 0$ and $\epsilon \to \infty$?

**Relation To Broader Scientific Literature:**

This paper addresses two topics of great interest to the ICML audience at large: sharp asymptotic characterizations of high-dimensional regression and adversarial training. It is well-situated within the literature, and (as I discuss below) I think the authors do a good job of referencing relevant prior work.

**Theoretical Claims:**

The main theorems are extensions of the results of Loureiro et al. 2021, and their proofs consist in no small part of quoting appropriate results from that earlier work. Based on my knowledge of Loureiro et al. 2021, I thus believe them to be correct.

---

> ### Author Rebuttal · Authors · 2025-04-01
>
> Thank you for the thoughtful and insightful review. Regarding all the typos and clarity suggestions, we will fix them in the camera-ready version.
>
> > [...] I think an experiment showing that the main conceptual result (i.e., regularizing with the dual norm improves performance) transfers to a real dataset would enhance the paper. [...]
>
> We refer to the answer given to Reviewer BhWC, where we explain how a similar behavior is observed for a classification task on 0 vs. 1 MNIST.
>
> > I wish the authors were able to extract more insight directly from Theorem 3.15. I fully appreciate that these self-consistent equations are not easily analyzable, but Remark 3.17 doesn't really say anything specific to this result. It merely restates the standard idea that the order parameters are sufficient statistics, and mentions the proof strategy (which is re-stated again in lines 297-301). Are there any special cases (beyond isotropic data) for which you could make something more of this result?
>
> The system of equations in the theorem could provide insights for specific questions. For instance, studying them perturbatively might help reveal how generalization error scales with model parameters. As an example, [Vilucchio et al. 2024] derived such scalings for a robust regression setting.
>
> However, the analytical progress in that paper relies critically on their choice of loss functions ($\ell_2$, $\ell_1$, and Huber loss) which have closed-form proximal operators, and on the regression setting they consider. In our binary classification context, the proximal operator lacks an analytical form for all the commonly used classification losses, making similar closed-form derivations significantly more challenging to obtain.
>
> > [...] I acknowledge that a complete analysis of the transition to optimality of the dual norm with increasing perturbation scale is out of reach, but (in the same vein as my previous comment) I wish the authors showed how more can be done with access to the sharp asymptotic. At the very least, the authors should show how loose the Rademacher complexity bounds are.
>
> Thank you for the suggestion. Indeed, it would be useful to show that the Rademacher complexity bounds are numerically loose. We will add in Appendix B the high-dimensional version of the Rademacher complexity bounds. For instance, see the [attached anonymized plot (bound_comparison.pdf)](https://anonymous.4open.science/r/imcl2025-rebuttals-940D) that shows that the $\ell_2$ bound $\left(\frac{ \mathcal{W}_2}{ \sqrt{\alpha}} \left(1 + \sqrt{\frac{1}{\lambda_1 (\Sigma) }} \right),\right)$ where $\lambda_1(\Sigma)$ is the smallest eigenvalue of the perturbation matrix $\Sigma$, is numerically loose (and even increases with $\alpha$). It corresponds to the blue lines shown in the Figure.
>
> > Given the smooth transition between $r^\star = 2$ and $r^\star = 1$ with increasing $\epsilon$ that you observe in Figure 4, can you gain any insight by expanding around $\epsilon = 0$ and $\epsilon \to \infty$?
>
> The primary challenge with formal expansions in this binary classification context is that the system of equations remains unsolvable and depends on integrals at any order in the expansion.
> As a result, the expansion would depend on a specific scaling ansatz in $\varepsilon$, where the coefficients at each order of the expansion are solutions of a similar system of equations as the original one. Thus, it appears to be not possible to obtain such an expansion without resorting to numerical solutions.

---

> > ### Comment · Reviewer_jKDA · 2025-04-01
> >
> > Thank you for your detailed response; my main concerns are addressed. I appreciate the difficulty of finding closed-form solutions, and the challenges of a perturbative expansion in $\epsilon$. I will maintain my score, as I am in favor of acceptance.

---

### Official Review · Reviewer_BhWC · 2025-03-12

**Overall Recommendation:** 3

**Summary:**

This work studies how to select the appropriate regularization norm in high-dimensional adversarial training for binary classification. The authors provide an exact asymptotic description of the robust, regularized empirical risk minimizer for various adversarial attacks and regularization norms. They also conduct a uniform convergence analysis, deriving bounds on the Rademacher Complexity for these problems. Using their theoretical findings, they quantitatively characterize the relationship between perturbation size and the optimal choice of regularization norm.

**Claims And Evidence:**

The results in this paper are rigorously proved.

**Essential References Not Discussed:**

It appears that the essential references are cited.

**Experimental Designs Or Analyses:**

The experimental designs appear to be reasonable. However, the algorithm used to solve the minimization problem (7) on page 3 is not clearly explained. Providing details about the algorithm used in the simulation studies would be helpful for understanding the approach and replicating the results.

**Methods And Evaluation Criteria:**

The evaluation methods used are generally appropriate for assessing the proposed method. However, it would also be interesting to test the robustness of the results by examining scenarios where the linear classifier assumption is violated. This would provide valuable insights into how the method performs under model misspecification.

**Other Comments Or Suggestions:**

No other comments.

**Other Strengths And Weaknesses:**

No additional comments.

**Questions For Authors:**

This paper focuses on binary classification models.
I wonder if the results can be extended to linear regression models and generalized linear models?

**Relation To Broader Scientific Literature:**

This work primarily focuses on linear classifiers for binary classification. While the results appear to be of theoretical interest, it is unclear how they apply to practical situations involving high-dimensional, complex data, where nonlinear classifiers are typically required. Explaining the implications of these findings in such contexts would provide valuable insights into their practical utility.

**Theoretical Claims:**

I have reviewed the proofs, and they appear to be correct.

---

> ### Author Rebuttal · Authors · 2025-04-01
>
> Thank you very much for your critical and thorough review of our work.
>
> > The evaluation methods used are generally appropriate for assessing the proposed method. However, it would also be interesting to test the robustness of the results by examining scenarios where the linear classifier assumption is violated. This would provide valuable insights into how the method performs under model misspecification.
>
> We agree that this is a nice addition for our paper and are adding such an experiment. We consider the case of 0 vs 1 MNIST classification. We take 0 and 1 MNIST images, normalize their pixel values in $[0,1]$ and flatten them to have a $d=784$ dimensional $\bf{x}_i$. The labels are converted to be $-1$ and $+1$. We then train as per eqs. (6,7) with this dataset, using subsets of a suitable size to fix various $\alpha = n / d$ and then test robust and clean error as per eqs. (3,4) with $1000$ new samples.
> We attach the results of two experiments with $\varepsilon = 1.0, 2.0$ in an [anonymized folder](https://anonymous.4open.science/r/imcl2025-rebuttals-940D).
> We see that by reducing the number of training samples (lowering $\alpha$) the robust error differs based on the choice of the regularization. The optimal choice of regularization is $r=1$, which is constistent with our idealized theoretical model.
> In the camera ready version, we will include a discussion of these results.
>
> > [...] the algorithm used to solve the minimization problem (7) on page 3 is not clearly explained. [...]
>
> Thank you for pointing this out. Equation (7) corresponds to a convex problem as explained in Appendix A (see eq (37)). Thus, it can be solved using a general purpose convex solver. We used the L-BFGS-B algorithm with random normal initialization and a stopping tolerance of `1e-5` for the gradient. We will include these experimental details in the revised version.
>
> > This paper focuses on binary classification models. I wonder if the results can be extended to linear regression models and generalized linear models?
>
> The key step that allows the analytical description of the problem is going from eq. (36) to eq. (37). This step relies on the fact that we consider binary classification, as explained in Appendix A, and cannot easily be generalized to more complicated classification tasks.

---

### Official Review · Reviewer_tJzf · 2025-03-17

**Overall Recommendation:** 4

**Summary:**

The authors investigate the impact of regularization geometry on adversarial training in a binary classification problem. The primary objective of the study is to control the robust generalization error under input perturbations constrained by a specified norm. To achieve this, they optimize the robust empirical (regularized) risk under a specified regularization. The work is presented in three key steps:

1. The authors derive an exact asymptotic characterization of the performance of Regularized Robust Empirical Risk Minimization (RERM) for various combinations of perturbation and regularization norms, including both $\ell_p$-norms and $\|\cdot\|_{\Sigma}$-norms.
2. They establish novel uniform convergence bounds using Rademacher complexity results for different norm geometries. They prescribe the dual norm of the perturbation norm to be used as regularizer for the linear classification problem.
3. They conduct synthetic experiments to show the validity of their theoretical results.

## update after rebuttal
I would like to thank the authors for their response. They have addressed my questions, and I have decided to increase my score.

**Claims And Evidence:**

Although I have not examined the mathematical details carefully, the claims and arguments presented in the main text appear to be valid.

**Essential References Not Discussed:**

The relevant references are discussed adequately.

**Experimental Designs Or Analyses:**

The work is mostly theoretical, although synthetic experiments have been presented which make sense for the problem setting.

**Methods And Evaluation Criteria:**

The work is mostly theoretical, although synthetic experiments have been presented which make sense for the problem setting.

**Other Comments Or Suggestions:**

Please see the previous sections.

**Other Strengths And Weaknesses:**

Although the binary classification problem setting appears simple, the exact asymptotic characterization of the performance of Regularized Robust Empirical Risk Minimization (RERM) and the novel uniform convergence bounds derived using new Rademacher complexity results are valuable contributions to the machine learning community.

To further strengthen the paper, it would be beneficial to discuss the practical applicability of the assumptions. For example:
1. Can Assumption 3.11 be extended to accommodate zero-mean sub-Gaussian covariates?
2. How restrictive is Assumption 3.13 in practice, and how often is it likely to hold in real-world scenarios?

**Questions For Authors:**

Please see the previous sections.

**Relation To Broader Scientific Literature:**

The contributions are well-situated within the context of existing literature. The work by Tanner et al., 2024 appears to be the closest to the proposed study. Specifically, the results in Section 3 bear resemblance to the techniques and findings presented by Tanner et al., 2024, though applied to $\ell_2$-regularization and perturbations in the $\|\cdot\|_{\Sigma}$-norm.

Could the authors provide a comparison and contrast of their results with those of Tanner et al., 2024, particularly in terms of the technical settings and innovations in their technical analysis? Additionally, what were the key technical challenges in extending the work of Tanner et al., 2024 to the proposed setting? Addressing these questions would clarify the novelty and technical contributions of this work.

**Theoretical Claims:**

I have not examined the mathematical details carefully.

---

> ### Author Rebuttal · Authors · 2025-04-01
>
> Thank you very much for your thorough review of our work.
>
> > Could the authors provide a comparison and contrast of their results with those of Tanner et al., 2024, particularly in terms of the technical settings and innovations in their technical analysis? Additionally, what were the key technical challenges in extending the work of Tanner et al., 2024 to the proposed setting?
>
> In a nutshell, Tanner et al. 2024 study Mahalanobis-norm perturbations and regularization with an $\ell_2$ penalty, while Section 3 in our paper presents results for (a) Mahalanobis-norm perturbations and general Mahalanobis-norm regularizations and (b) $\ell_p$ norms for both perturbation and regularization.
> On the technical side the proofs that appear in [Tanner et al. 2024] are based on a mapping to a Generalised Approximate Message Passing algorithm.
> Our results are proven via Gordon's Theorem (Theorem A.1) and allow for a broader range of loss and regularization functions.
>
> > Can Assumption 3.11 be extended to accommodate zero-mean sub-Gaussian covariates?
>
> This is an excellent question about practical extensions. While our theoretical guarantees rely on Gaussian assumptions for applying Gordon's Theorem, recent work on universality [Montanari & Saeed (2022), and Dandi et al. (2023)] suggests these results often extend to more general distributions including sub-Gaussian covariates.
>
> The key technical challenge for a formal extension would involve using more general concentration inequalities for sub-Gaussian random variables [Vershynin, "High-Dimensional Probability", 2018]. However, maintaining the freedom of chosing an arbitrary non-increasing convex loss function would require additional technical machinery beyond the scope of this initial work. We believe this is a promising direction for future research.
>
> > How restrictive is Assumption 3.13 in practice, and how often is it likely to hold in real-world scenarios?
>
> Assumption 3.13 (simultaneous diagonalizability) is primarily a technical assumption that allows for a well specified setting. In practice, this assumption is less restrictive than it might appear.
> For example, this property holds when the matrices $\Sigma_w, \Sigma_\delta$ share principal components, which occurs naturally in many signal processing applications where the same underlying factors drive both input correlations and noise structure.
> The assumption tries to mimic PCA applied to data, which is common practice in many ML pipelines. In such cases, the perturbation and regularization matrices would indeed share eigenvectors with the transformed data covariance.
>
> Nonetheless we agree that future work could relax this assumption to broaden applicability, and our Rademacher complexity analysis in Section 4 already takes a step in this direction by providing distribution-agnostic guarantees.

---

### Decision · Program_Chairs · 2025-05-01

**Decision:**

Reject

**Comment:**

The paper studies regularization in adversarial training (with epsilon balls) for binary classification. Rademacher complexity is used to obtain generalization bounds for various norms. Experiments were also conducted.

While the reviewers were supportive of this paper, none of the reviewers strongly argued for acceptance. Thus I consider this a borderline paper, but I finally decided for rejection. Several aspects for improvement were mentioned including for instance relaxing some assumptions (from Gaussian to sub-Gaussian), discussing the practicality of some other assumptions (convergence of spectra), discuss multi-class extensions, among others.